# Spatiotemporal changes of seismicity rate during earthquakes

Chieh-Hung Chen[1,2*], Yang-Yi Sun[2], Strong Wen[3], Peng Han[4], Li-Ching Lin[5], Huai-Zhong Yu[6], XueMin Zhang[7], Yongxin Gao[8], Chi-Chia Tang[1,2], Cheng-Horng Lin[9], Jann-Yenq Liu[10,11,12]

1. State Key Laboratory of Geological Processes and Mineral Resources, China University of Geosciences, Wuhan, China
2. Institute of Geophysics and Geomatics, China University of Geosciences, Wuhan, China
3. Department of Earth and Environmental Sciences, National Chung Cheng University, Chiayi, Taiwan
4. Department of Earth and Space Sciences, Southern University of Science and Technology, Shenzhen, China
5. Department of System Engineering and Naval Architecture, National Taiwan Ocean University, Keelung, Taiwan
6. China Earthquake Networks Center, Beijing, China
7. Institute of Earthquake Forecasting, China Earthquake Administration, Beijing, China
8. School of Civil Engineering, Hefei University of Technology, Hefei, China
9. Institute of Earth Sciences, Academia Sinica, Taipei, Taiwan
10. Center for Astronautical Physics and Engineering, National Central University, Taoyuan, Taiwan
11. Department of space science and engineering, National Central University, Taoyuan, Taiwan
12. Center for Space and Remote Sensing Research, National Central University, Taoyuan, Taiwan

**\* Corresponding Author:**
Chieh-Hung Chen, E-mail: nononochchen@gmail.com
Institute of Geophysics and Geomatics,
China University of Geosciences, Wuhan, Hubei, 430074, China

**Abstract**

36        Scientists demystify stress changes within tens of days before a mainshock and

often utilize its foreshocks as an indicator.   Typically, foreshocks are detected near
fault zones, which may be due to the distribution of seismometers.   This study
investigates changes in seismicity far from mainshocks by examining tens of thousands
of $M \geq 2$ quakes that were monitored by dense seismic arrays for more than 10 years in
Taiwan and Japan.   The quakes occurred within epicentral distances ranging from 0
km to 400 km during a period of 60 days before and after the mainshocks that are
utilized to exhibit common behaviors of seismicity in the spatiotemporal domain.   The
superimposition results show that wide areas exhibit increased seismicity associated
with mainshocks being more than several times to areas of the fault rupture.   The
seismicity increase initially concentrates in the fault zones, and gradually expands
outward to over 50 km away from the epicenters approximately 40 days before the
mainshocks.   The seismicity increases more rapidly around the fault zones
approximately 20 days before the mainshocks.   The stressed crust triggers ground
vibrations at frequencies varying from $\sim 5 \times 10^{-4}$ Hz to $\sim 10^{-3}$ Hz (i.e., variable frequency)
along with earthquake-related stress that migrates from exterior areas to approach the
fault zones.   The variable frequency is determined by the observation of continuous
seismic waveforms through the superimposition processes and is further supported by
the resonant frequency model.   These results suggest that the variable frequency of
ground vibrations is a function of areas with increased seismicity leading to earthquakes.

Keywords: foreshocks; resonance frequency; earthquake-related stressed area

## 1. Introduction

Numerous studies (Reasenberg, 1999; Scholz, 2002; Vidale et al., 2001; Ellsworth and Beroza, 1995) reported that foreshocks occur near a fault zone and migrate toward the hypocenter of a mainshock before its occurrence. The spatiotemporal evolution of foreshocks is generally considered to be an essential indicator that reveals variations in earthquake-related stress a couple of days before mainshocks. After detecting these variations, scientists installed multiple instruments along both sides of the fault over short distances to monitor the activity of the fault. However, these instruments typically detect small vibrations near the fault zone. Stress accumulates in a local region near a hypocenter triggering earthquake occurrence that is concluded from the sparse distribution of seismometers.

Bedford et al. (2020) analyzed the GNSS data and observed crustal deformation in a thousand-kilometer-scale area before the great earthquakes in the subduction zones. Chen et al. (2011, 2014, 2020a, 2020b) filtered the crustal displacements before earthquakes using the GNSS data through the Hilbert-Huang transform. The filtered crustal displacements in a hundred(thousand)-kilometer-scale area before the moderate-large (M9 Tohoku-Oki) earthquakes exhibit paralleling azimuths that yield an agreement with the most compressive axes of the forthcoming earthquakes (Chen et al., 2014). On the other hand, Dobrovolsky (1979) estimated the size of the earthquake preparation zone using the numerical simulation method and found that the radius (R) of the zone is proportional to earthquake magnitude (M). In addition, the relationship can be written by using a formula of $R=10^{0.43M}$. These results suggest that a stressed area before earthquakes is obviously larger than the rupture of fault zones. However, it is a big challenge to monitor stress changes in a wide area beneath the ground. A simple way to imagine this is if we place a stick on a table, then hold and try to break the stick. The stress we making on the stick can apply to either a limited local region or to both ends of it. Migrations and propagations of the loading force can be detected according to the changes of strain and the occurrence of microcracks. This common sense suggests that the spatiotemporal evolution of earthquake-related stress appearing

a couple of days before mainshocks can be recognized if we can trace the occurrence
of relatively-small quakes in a wide area (Kawamura et al., 2014; Wen and Chen, 2017).
Here we take advantage of earthquake catalogs obtained by dense seismic arrays in
Taiwan and Japan to expose foreshocks distributing over a wide area instead of a local
region.

**2. Methodology**
The ability to detect relatively-small quakes depends on the spatial density and
capability of seismometers.   Taiwan and Japan are both the most famous high-
seismicity areas in the world.   Dense seismometers evenly distributed throughout the
whole area are beneficial for monitoring the earthquake occurrences near to and far
away from fault zones (Chang, 2014).   Earthquake catalogs retrieved from Taiwan and
Japan were obtained from the Central Weather Bureau (CWB), Taiwan and the Japan
Meteorological Agency (JMA), respectively.   To distinguish dependencies from
independent seismicity, the earthquake catalogs are declustered.   Therefore, the
ZMAP software package for MATLAB (Weimer, 2001) was utilized to remove and/or
omit influence from duplicate events, such as aftershocks.   The declustering algorithm
used in ZMAP is based on the algorithm developed by Reasenberg (Reasenberg, 1985).
We classify clusters by using the standard input parameters (proposed in Reasenberg,
1985 and Uhrhammer, 1986) for the declustering algorithm.   Because the aftershock
clusters in a small area and in a short period of time do not conform to the Poisson
distribution, which requires removing the aftershocks from the earthquake sequence.
Therefore, some parameters can be set as follow: The look-ahead time for un-clustered
events is in one day, and the maximum look-ahead time for clustered events is in 10
days.   The measure of probability to detect the next event in the earthquake sequence
is 0.95.   The effective minimum magnitude cut-off for the catalog is given by 1.5, and
the interaction radius of dependent events is given by 10 km (van Stiphout et al., 2012).
Earthquakes with depth > 30 km were eliminated from the declustered catalogs to
understand seismicity changes before mainshocks mainly in the crust.
Before the analytical processes in this study, we assumed that earthquakes with
relatively-small magnitude can be the cracks and potentially related to the far
mainshocks based on the large seismogenic areas (Bedford et al., 2020).    The
minimum magnitudes of completeness Mc are 2.0 and 0.0 that can be determined by
the declustered earthquake catalogs in Taiwan and Japan, respectively (also see Figs.
S1–S4).    The earthquakes with M ≥ 2 are selected and utilized in this study for fair
comparison of the seismicity changes during earthquakes in Taiwan and Japan.    We
classified the selected earthquakes via their magnitudes into three groups (i.e., 3 ≤ M <
4, 4 ≤ M < 5 and 5 ≤ M < 6).    Note that the classified earthquakes in each group are
determined as the break events (i.e., the mainshocks).    In contrast, the other selected
earthquakes with magnitudes smaller than the minima of the classified magnitude are
determined as the crack events.
We construct a spatiotemporal distribution of the crack events for each break quake.
The spatiotemporal distribution from 0 km to 400 km away from the epicenter of the
break quake during a period of 60 days before and after the break occurrence is
constructed to illustrate the relationship between the crack events and the break quake
in the spatial and temporal domain.    Note that the spatial and temporal resolutions of
the grids of the spatiotemporal distribution are 10 km and 1 day, respectively, based on
the declustering parameters in the ZMAP software (Weimer, 2001).    We count the
crack events in each spatiotemporal grid according to distance away from the epicenter
and the differences in time before and after the occurrence of the break quake.
The superimposition process, a statistical tool utilized in data analysis, is capable
of either detecting periodicities within a time sequence or revealing a correlation
between more than two data sequences (Chree, 1913).    The process is known as the
superposed epoch analysis (Adams et al., 2003; Hocke, 2008).    In practice, the
superimposition is a process to stack numerous datasets that can migrate unique features
for a few datasets and enhance common characteristics for the most datasets.    The
count in each grid of the spatiotemporal distributions for all the break quakes are
superimposed as a total one based on the occurrence time and epicentral distance of the
break quakes.   The total count of the superimposed distribution in each spatiotemporal
grid is normalized to seismic density (count/km$^2$) for comparing to the total number of
the break quakes and the related spatial area.   Moreover, we compute the average
values every distance grid using the seismic densities 60 days before and after the quake.
The average values are subtracted from the seismic densities and the obtained
differences are divided by the average values in each distance grid to obtain the
normalized variation clarifying changes of the seismic density in the spatiotemporal
domain.

**3. Analytical results**
The earthquakes with magnitude ≥ 2 listed in the declustered catalogs of Taiwan
from January 1991 to June 2017 are utilized to construct a spatiotemporal distribution
of foreshocks and aftershocks corresponding to the quakes with $3 \leq M < 4$.   We
superimposed all the crack events corresponding to the 15625 quakes ($3 \leq M < 4$).
The seismic density is more than 1000 times greater in a hot region at a distance of 10
km away from an epicenter (which is generally considered to be the gestation area of
foreshocks) than it is in areas located > 200 km from the epicenter (Fig. 1a).   The
sudden increase of seismic density suggests that earthquake-related stress accumulates
mainly around the hot region, triggering many foreshocks a few days before the
earthquakes with $3 \leq M < 4$.   This partial agreement of the numerous recent studies
reported that the seismicity migrates toward the fault rupture zone within tens of
kilometers from epicenters a couple of days before earthquakes (Kato et al., 2012, Kato
and Obara, 2014; Liu et al., 2019).   Meanwhile, the events mainly occur 0–1 day after
the quakes that is irrelevant to the smaller distribution 0–1 day before the quakes (also
see Fig. 1).   The seismic density close to epicenters (Fig. 1) suddenly increases before
and gradually decreases after the quakes.   The irrelevance and the differences of
changes rates with epicentral distance smaller than 20 km before and after the quakes
reveal that the increase of seismicity before the quakes is not contributed by the
seismicity after due to the analytical processes in this study.   In addition, these
analytical results of the seismic activity are also in agreement with the studies in
Lippiello et al. (2012, 2017, 2019) and de Arcangelis et al. (2016) regard for distinct
methods.

178         On the other hand, the increase of seismic density is not only always limited within

the hot region, but also extends outward to a distance of over 50 km away from the
epicenters about 0–40 days leading up to the occurrence of the quakes (Fig. 1a). We
further examine the spatiotemporal changes in the seismic density up to the M ≥ 4
quakes utilizing the same superimposition process (Figs. 1b–c). The expansion of the
increased seismic density about 0–40 days leading up to the occurrence of the quakes
and the sharp increases of seismic density a few days before the quakes that can be
consistently observed using the M ≥ 4 quakes in Figs. 1b–c. Similar results (i.e., the
sharp increases of seismic density a few days before the quakes and areas where the
increase of the seismicity density is much larger than that of the hot region) can also be
obtained using the earthquake catalogs between 2001 and 2010 from the Japan
Meteorological Agency (JMA) in Japan (Figs. 1d–f). Note that the earthquakes that
occurred in the northern side of the latitude of 32°N were selected from the Japan
catalogs. The selection is based on that the earthquakes occurred in the area monitored
by the dense seismometer network and to avoid the double count of events in the
Taiwan catalogs. The normalized variations correspond to seismic density in Fig. 1
are shown in Fig. 2. The radii of the positive normalized variations are approximately
50 km while earthquake magnitude increases from 3 to 6 in Taiwan (Figs. 2a–c). The
land area of Taiwan is approximately 250 km by 400 km, which causes underestimation
of the seismic density in the spatial domain. In contrast, the positive normalized
variations roughly expand along the radii ranging from 50 km to 150 km, while
earthquake magnitude increases from 3 to 6 in Japan (Figs. 2d–f). However,
variations in the lead time mostly range from 40 days to 20 days, and relationships
between the positive normalized variations and the earthquake magnitude can be found
neither in Taiwan nor Japan (Fig. 2).

203         In short, the expansion of the increase of seismic density becomes mitigation and

may no longer be impact a place at distances > 200 km away from the epicenters for
the earthquakes with magnitude < 6.    The increase of seismicity density before the
quakes suggests that the accumulation of the earthquake-related stress in the crust
originates from the hot region, and gradually extends to an external place before
earthquakes occur.    The area of this external place is several times that of a fault
rupture zone that is concluded based on the sparse seismic arrays of the past.    If a
quake can excite seismicity changes over a wide area (i.e., over 50 km by 50 km), any
crustal vibration related to stress accumulation before earthquakes can be too small to
be identified from continuous seismic waveforms at one station.    In contrast, crustal
vibrations can be a common characteristic of continuous seismic waveforms at most
stations around fault zones due to that seismicity changes dominated by earthquake-
related stress accumulation distributes in a wide area.

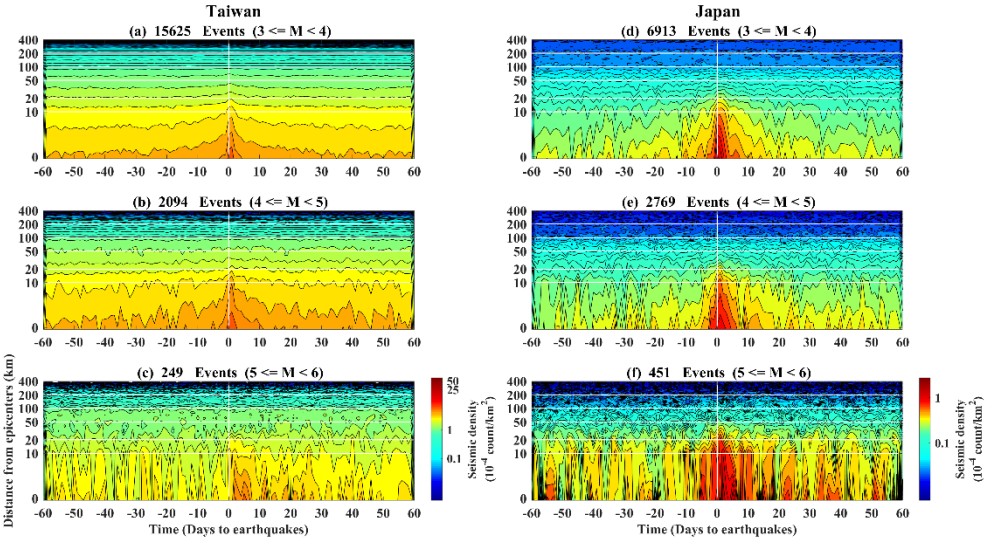



Fig. 1.    Spatiotemporal seismic density distributions in Taiwan and Japan.    The
seismic densities constructed by using the declustered earthquake catalogs of Taiwan
and Japan are shown in the left and right panels, respectively.    The seismic density
reveals changes in seismicity at distances from the epicenters ranging from 0 km to 400
km at up to 60 days before and after quakes in a particular magnitude group.    The
superimposed number in each grid is further normalized for a fair comparison by using
the total number of quakes and their areas.    Notably, the total number of quakes is
shown in the title of each diagram.

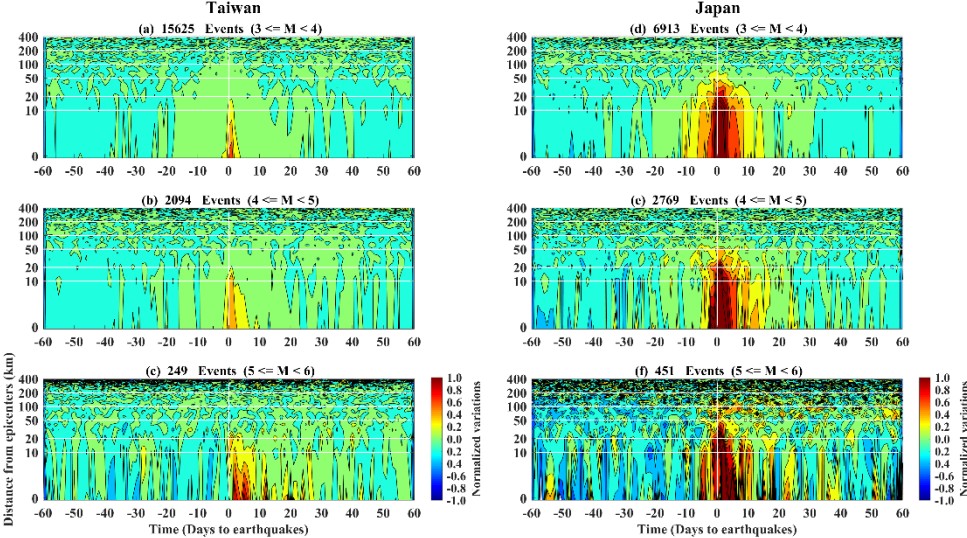



Fig. 2.    Changes of the normalized spatiotemporal variations in Taiwan and Japan.
The normalized variations correspond to the seismic density in Taiwan and Japan (in
Fig. 1) are shown in the left and right panels, respectively.    The colors reveal changes
of the normalized variations at distances from the epicenters ranging from 0 km to 400
km at up to 60 days before and after quakes in a particular magnitude group.

**4.  The principal component analysis (PCA) on the continuous seismic waveforms**

Seismic waveforms obtained from 33 broadband seismometers operated by
National Center for Research on Earthquake Engineering (NCREE) of Taiwan, within
a temporal span of approximately one year (from June 2015 to June 2016) are utilized
in this study.    Note that two seismometers of them are eliminated from following the
analytical processes due to long data gaps.    The principal component analysis (PCA)
method (Jolliffe, 2002) is utilized to retrieve the possible stress-related common signals
from continuous seismic waveforms on the vertical component at thirty-one seismic
stations over a wide area and to mitigate local noise simultaneously.    Fig. 3a shows
that the energy and the cumulative energy of the principal components derived from the
continuous seismic waveforms at the 31 stations. The energy of the first principal
component is about 12% that is more than 3 times to the following ones. Thus, we
determined the first principal component to be the common signals of the ground
vibrations before earthquakes. Fig. 3b reveals changes in the common signals during
the study period along the time. However, no obvious changes can be observed in the
temporal domain.

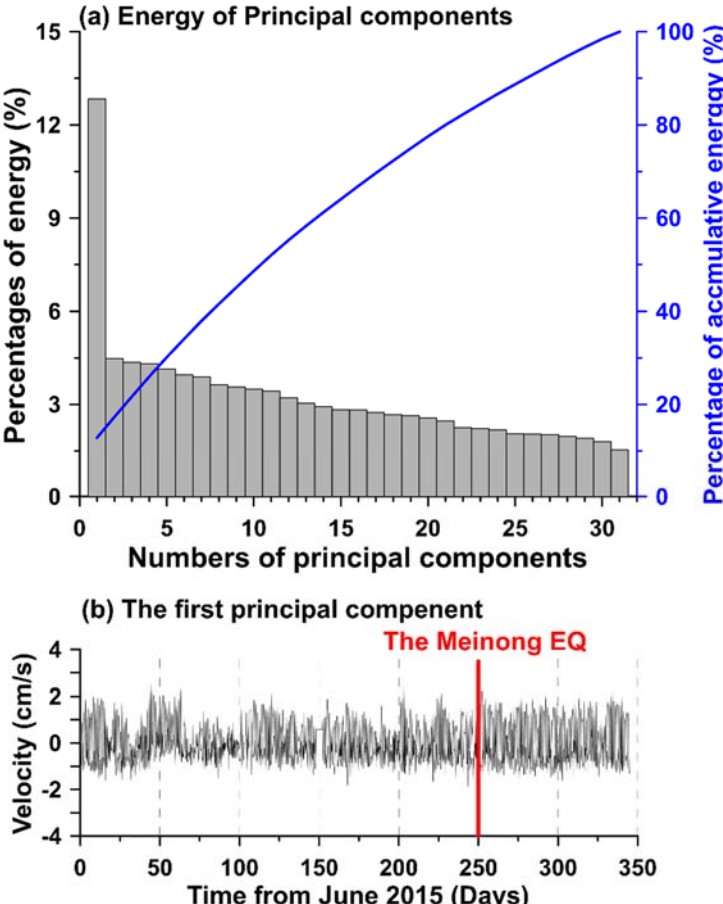


Fig. 3. The energy and the first principal component derived from vertical seismic
velocity data from the 31 stations. The energy and the cumulative energy of the
principal components are shown in (a). Bars denote the energy of each principal
component. The blue line shows the variation of the cumulative energy from distinct
used principal components. The variations of the first principal component during the
period (i.e., from June 2015 to June 2016) are revealed in (b). The red vertical line
indicates the occurrence time of the M6.6 Meinong earthquake (on February 2, 2016).

Thus, we sliced the common signals into several time spans using a 5-day moving
window with one-day steps to show time-varying changes.   The common signals in
each time span are transferred into the frequency domain using the Fourier transform
to investigate frequency characteristics of ground vibrations before earthquakes.   The
amplitudes are normalized using the frequency-dependent average values computed
from the amplitude 30 days before and after earthquakes via the temporal division.
Here, we take the M6.6 Meinong earthquake (Wen and Chen, 2017, Chen et al., 2020c)
as an example to understand the changes of the amplitude of the common signals in the
spatiotemporal domain (Fig. 4a).   Distinct patterns in the amplitude-frequency
distributions can obviously be observed before and after the earthquake at frequency
higher than $5 \times 10^{-4}$ Hz (also see Figs. 4e and 4f).   The amplitude at the frequency close
to $5 \times 10^{-4}$ Hz was obviously enhanced approximately 20–40 days before the earthquake.
Hereafter, the enhancements were significantly reduced and reached to a relatively-
small value a few days after the earthquake.   Meanwhile, the frequency is close to
$2 \times 10^{-4}$ Hz approximately 60 days before the earthquake and tends to be high near $10^{-3}$
Hz a few days before the event (also see Figs. 4e–4f).   We next superimpose the
amplitude based on the occurrence time of the 17 earthquakes with $4 \leq M < 5$ and the
109 earthquakes with $3 \leq M < 4$ during the one-year temporal span shown in Figs. 4b
and 4c, respectively.   The consistent variations (i.e., the frequency is close to $2 \times 10^{-4}$
Hz approximately some days before the quakes tending to be high near $10^{-3}$ Hz a few
days before the quakes) that can be observed in Figs. 4b and 4c.
Here, we retrieve the ratios at three frequencies of approximately $1 \times 10^{-4}$ Hz, $5 \times 10^{-4}$
Hz, and $1 \times 10^{-3}$ Hz to reveal the relationship between the enhancements and
earthquake magnitudes (Figs. 4d–4f).   For the Meinong earthquake, the enhancements
could be identified at the low frequency of approximately $1 \times 10^{-4}$ Hz.   The ratios
exhibit a relatively-large value of ~1.2 about 90 days earlier than the earthquake (Fig.
4d).   The ratios rapidly decrease to a relatively-small value of ~0.5 near 60 days before
the earthquake.   The enhancements with the maxima reach ~1.6 appeared ~30 days
before the earthquake.   After the earthquake, the ratios fluctuate and recover as a
relatively-large value of ~1.2 about 100 days later than the earthquake. Regarding
earthquakes with relatively-small magnitude, the enhancements at $1 \times 10^{-4}$ Hz is ~1.2 for
the group of $4 \leq M < 5$, and ~1.1 for the group of $3 \leq M < 4$ between 30 days and 50
days before the earthquake occurrence (Fig. 4d). Similarly, the enhancements at $5 \times 10^{-4}$
Hz is ~1.4 for the Meinong earthquake, ~1.15 for the group of $4 \leq M < 5$, and ~1.05
for the group of $3 \leq M < 4$ between 5 days and 30 days before the earthquake occurrence
(Figs. 4e). The enhancements at $1 \times 10^{-3}$ Hz is ~1.15 for the Meinong earthquake,
~1.15 for the group of $4 \leq M < 5$, and ~1.05 for the group of $3 \leq M < 4$ between 2 days
and 30 days before the earthquake occurrence (Fig. 4f). The ratios at the three
frequencies in Figs. 4d–4f suggest that the amplitude ratios of the enhancements and
earthquake magnitudes generally show a proportional relationship. However, the
ratios at $1 \times 10^{-3}$ Hz with a relatively-large value of ~1.6 can be observed during the
period of 60–45 days before the Meinong earthquake due to unknown disturbances (Fig.
4f).
The findings suggest that the common-mode ground vibrations exist in a wide area
before earthquakes due to the signals being retrieved from the most stations distributing
the whole Taiwan island through the PCA method. In short, the common-mode
vibrations are very difficult to be identified from the time-series data but become
significant in the frequency domain. If the expansion of the seismoeneric areas and
the existence of the common-mode ground vibrations are true, the next step is to
determine the potential mechanism hidden behind this nature.

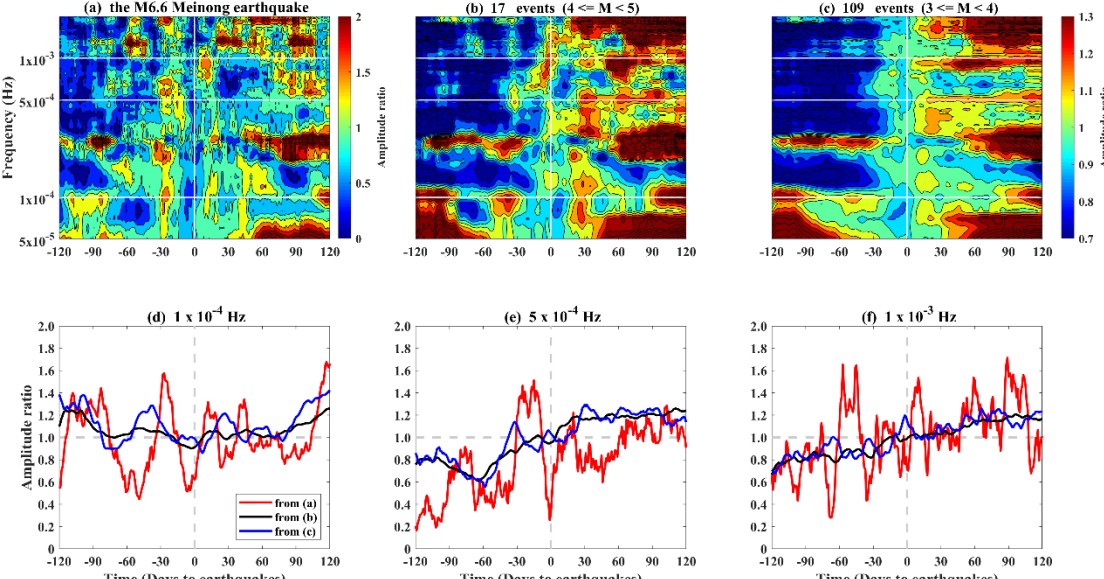

Fig. 4. The amplitude ratio of the superimposed time-frequency-amplitude distribution associated with earthquakes with distinct magnitudes. The superimposed results 120 days before and after quakes with the M6.6 Meinong earthquake, $4 \leq M < 5$ and $3 \leq M < 4$ are shown in (a), (b) and (c), respectively. The distribution is normalized for comparison by using the average amplitude in each frequency band of 30 days before and after the quakes. The total number of earthquakes in each magnitude group is shown in the title of each diagram. Variations of the amplitude ratios in (a)–(c) at frequencies of about $1 \times 10^{-4}$ Hz, $5 \times 10^{-4}$ Hz, and $1 \times 10^{-3}$ Hz during the same period are shown in (d), (e) and (f), respectively.

## 5. Discussions

Walczak et al. (2017) repeatedly observed stressed rocks exciting long-period vibrations during rock mechanics experiments. Leissa (1969) reported that the resonance frequency of an object is proportional to its Young's modulus and exhibits an inverse relationship to its mass. Based on the crust, the outermost of the Earth, is lamellar, we assume that the earthquake-related stress accumulates in the volume of a square sheet with a width of 100 km, which is determined by using a distance of 50 km away from an earthquake due to the significant increase of the seismic density (Figs. 1 and 2). The resonance frequency near $3 \times 10^{-4}$ Hz (Fig. 4) can be derived from the

square sheet once the thickness of the volume is ranged between 500 meters and 1000 meters (Fig. S5).   Although we do not fully understand the causal mechanism of the thickness, the agreement with the spatiotemporal domain of the relatively-small quakes from the earthquake catalogs, the superimposition results of continuous seismic waveforms and the resonance frequency models suggest that the phenomenon of variable frequency may exist tens of days before earthquake occurrence and can be retrieved by broadband seismometers.

In this study, we determined the seismogenic areas using the relatively-small earthquakes in the spatiotemporal distribution and found that the areas are significantly larger than the fault rupture zone (Figs. 1 and 2).   Meanwhile, the ground vibrations can exhibit frequency-dependent characteristics at about $10^{-4}$ Hz (Fig. 4) that could relate to the large seismogenic areas due to the resonance model (Fig. S5).   If these are true, the seismo-TEC (total electron content) anomalies in the ionosphere, which is generally observed in a large-scale area with more than ten thousand square kilometers (Liu et al., 2009), are high potential to be driven by upward propagation of acoustic waves before earthquakes (Molchanov et al., 1998, 2011; Korepanov et al., 2009; Hayakawa et al., 2010, 2011; Sun et al., 2011; Oyama et al., 2016).   The existence of the ground vibrations can generate the acoustic-gravity waves that have been reported (Liu et al., 2016, 2017).   However, the acoustic-gravity waves in a period of < 300 seconds are difficult to propagate upward into the atmosphere and the ionosphere (Yeh and Liu, 1974; Azeem et al., 2018).   The wide seismogenic areas observed in this study can contribute the larger-scale ground vibrations at approximately $5\times10^{-4}$–$10^{-3}$ Hz that cover the frequency channel (< 1/300 Hz) for the acoustic-gravity waves propagating into the atmosphere and changing the TEC in the ionosphere.   Meanwhile, the seismo-atmospheric and the seismo-ionospheric anomalies in a large-scale area can also be supported by the acoustic-gravity waves due to the wide seismogenic areas. While partial aforementioned relationships cannot be quickly proven, the ground vibrations at a low frequency (< 1/300 Hz) in a wide area assist our understanding of the essence of the seismo-anomalies in the atmosphere and the ionosphere.

## 6. Conclusion

The process of stress migration in the spatiotemporal domain can be concluded from tracing the increase of seismicity according to the 10-year earthquake catalogs from dense seismic arrays in Taiwan and Japan. Areas with the increase of seismicity, where stress accumulates in the crust triggering earthquakes are serious underestimation using a sparse seismic array. Seismicity initially increases around hypocenters, and this can be observed more than 50 days before quakes through superimposing large numbers of earthquakes. The seismicity gradually increases along with the expansion of areas from fault zones to an area widely covering an epicentral distance close to 50 km approximately 20–40 days before earthquakes. The crustal resonance exists at a frequency near $5\times10^{-4}$ Hz when the expansion becomes insignificant. Instead of the spatial expansion, the sharp increase of seismicity around the hot regions suggests stress accumulation in fault zones generating crustal resonance at a frequency of up to $\sim10^{-3}$ Hz in the few days before earthquakes. Most broadband seismometers can observe the variable frequency of ground vibrations in Taiwan due to the comprehensive spatial coverage of resonant signals. The variable frequency depends on various stress-dominant areas that can be supported by the potential crustal resonance model. Seismic arrays comprise dense seismometers with a wide coverage are beneficial for monitoring the comprehensive process of stress migration in the spatiotemporal domain leading up to a faraway and forthcoming mainshock.

*Acknowledgements.* The authors appreciate scientists who devote to maintain instruments in the field and data centers in the office that leads chances to expose such interesting geophysical phenomena and understand potential processes during seismogenic periods. This research was funded by National Key R&D Program of China, grant number 2018YFC1503705; National Natural Science Foundation of China (Grants No. 41474038 and 41774048); the Spark Program of Earthquake Science of China (Grant No. xh17045); Ministry of Science and Technology of Taiwan (Grants No.

MOST 106-2116-M-194-016- and MOST 106-2628-M-008-002), and Sichuan earthquake Agency-Research Team of GNSS based geodetic tectonophysics and mantle-crust dynamics of Chuan-Dian region (Grant No. 201803). Meanwhile, this work was also supported by the Center for Astronautical Physics and Engineering (CAPE) from the Featured Area Research Center program within the framework of Higher Education Sprout Project by the Ministry of Education (MOE) in Taiwan.

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

**Data available**

The earthquake catalogs of Taiwan and Japan were obtained from the Central Weather Bureau (https://www.cwb.gov.tw/), and the Japan Meteorological Agency (JMA; https://www.jma.go.jp/jma/indexe.html), respectively. Seismic waveform data in Taiwan were provided by the Seismic Array of NCREE in Taiwan (SANTA; https://www.ncree.narl.org.tw/; please find the bottom for the English version in the top right side). The downsampled seismic waveforms with the temporal interval of 10 seconds can be utilized to reproduce the analytical results in this study through the MATLAB software that can be download at https://doi.org/10.5061/dryad.1jwstqjqq.

**Author contribution**

Y.Y.S. contributed discussion and revision; S.W. contributed discussion and revision; P.H. contributed data collection; L.C.L. contributed discussion and revision; H.Z.Y. contributed discussion; X.Z. contributed discussion; Y.G. contributed discussion; C.C.T. contributed discussion and revision; C.H.L. contributed discussion and revision; J.Y.L. contributed discussion and revision.

**Competing interests**

The authors declare that they have no known competing financial interests or personal

relationships that could have appeared to influence the work reported in this paper.