# Peer review of "Spatiotemporal changes of seismicity rate during"

_Natural Hazards and Earth System Sciences, 2020_

## Referee Comment (RC1) · Anonymous Referee #1 · 17 Apr 2020

The authors of the submitted research analyse with mathematical/statistical tools published seismic event catalogues from areas of high seismicity (Taiwan and Japan) in an attempt to identify patterns in the distribution on time and space of foreshocks of larger events. The presented results point to a distribution much wider of the foreshocks in time (up to 60 days) and space (up to 400 km of the main shock epicentre) of those currently accepted, even for main shocks of moderate magnitude. Such kind of analysis is promising; but I think as performed and presented in the submitted research is not yet ready for publication.

To me, it looks like the pieces of the submitted paper have been assembled in a hurry. The used methodologies need more explanation (why and how they are applied). Even more comments on the choice of the data are also needed. Moreover, a revision of the English syntax is needed. The sense of phrases is difficult to follow in many cases.

[Figure]

For these reasons I think the submitted research needs a deep and throughout revision before it can be accepted for publication.

In the following paragraphs I point some specific questions to be addressed on the submitted text.

-Methodology- Line 85. Citation Chen (2014) is not in the reference list. Lines 87-89. It is necessary to introduce a minimum description on how ZMAP software removes after-shocks. Lines 89-95. Idem: a minimum description on how clusters are classified and the meaning of the input parameters is necessary. Line 95. "The 10 of crack radii. . ." Do you mean 10 times the crack radii? Please, make clear this phrase. Line 96. Cite Stiphout (2012) is missing in the reference list. Lines 99-102. If I understand properly "crack" and "break" events are definitions you introduced in your analysis, being "crack events" quite equivalent to foreshocks and aftershocks. Please, make clear all these terms. Lines 102-104. There is some problem with the minimum completeness magnitude of the catalogues. Looking at figures S1-S4 it looks like the events in the Taiwan catalogue are included in the Japanese catalogue. Something should be said about this fact. Moreover, the Japanese catalogue comprises many events far away from the main islands (23-34N, 138-147E). I think this whole region does not have the dense seismometer network claimed in lines 83-85. All these points should be clarified in the text. Lines 109-110. I assume the spatial and temporal resolutions of the grid are a choice of the authors. If so you may comment if you try other resolutions and/or the reasons for your choice. Lines 113-116. The superimposition process statistical tool should be described. It is not a common tool in seismicity studies. Lines 118-121. It is not clear to me what "migrate rare characteristics" means. Please clarify this phrase.

-Analytical Results- Lines 130-132. All M2 events are foreshocks or aftershock of M3 events? Cannot they be independent events? Lines 132-134. What does it means that S/N ratio increases 135 times? Please clarify. Another issue: 17993 M3 events in the period 1991-2017/6 mean 2 events per day roughly. As Taiwan is 400 km long approx., it means that in a period of 60 days and 400 km as you are using in your analysis

there are many M3 earthquakes (100 approx.). It is not clear to me how the M2 events are associated with the M3 events. Maybe a good description of the superimposition process as applied in this case clarifies this issue. Lines 145-164. The previous pointed issues make difficult to follow the discussion on the results.

-Discussion- In fact this section presents a different analysis, using seismograms and the PCA method. Certainly, the presented analysis has been inspired by the results obtained in the previous section; but it can be performed and presented in a totally independent form. Thus, it should be better presented as another section of analysis results. It is not clear how you are using the PCA analysis in this case. Some figure showing an example of the procedure described on lines 217-222 can help. Lines 237-246. There are a lot of suppositions on the used dimensions. If horizontal dimensions (100 x 100 km2) can be roughly deduced/assumed from the previous results (obtained in this section and the previous one), the thickness between 500-1000 m needs a good explanation. Lines 275-276. I cannot see the need for this citation here. Even more, I has been unable to find the value 2700 km/m3 in the cited paper or on the additional information.

---

## Short Comment (SC1) · 10 Jun 2020

The manuscript presents results which, in my opinion, can be very relevant for the forecasting challenge. However I find that they are not well presented and the discussion appears quite confusing for the following reasons: 1) The first part of the manuscript is devoted to study spatio-temporal patterns of seismic activity before and after events in a given magnitude range, for Taiwan and Japan. There are many papers which report a similar increase of seismic activity before large earthquakes. The key point is if the observed increase can have a prognostic value or it can be explained within normal aftershock triggering. I just suggest some papers where this point is detailed discussed but the authors can find many other references therein (Lippiello et al., Scientific Reports 2012, de Arcangelis et al. Physics Reports 2016, Lippiello et al., Pure and Applied Geophys. 2017, Lippiello et al., Entropy 2019). In my opinion many of the results of sec.3 are not really interesting since they are probably artifact of the adopted

stacking procedure. Furthermore they are not strictly related to what for me are the main findings (see my point 2). Therefore, I believe that this section can be moved to the supplementary materials whereas in the main-text the authors can just summarize some results and discussing recent literature on this specific point. 2) Conversely, I strongly encourage the authors to move fig.S5 from the supplementary to the main-text. I am really impressed by this figure. In particular I find striking the result of the left panel which, if i correctly understand, is for a single M6.6 mainshock and therefore is not contaminated by spurious effects caused by the stacking procedure. This figure shows a change in the dominant frequency from roughly $10^{-4}$Hz up to 30 days before, to a much larger value before the mainshock. What I find really interesting is the analysis at a fixed frequency (around $10^{-4}$Hz) as function of the time from the mainshock. In this case you find that the mainshock occurrence time is a minimum" point in the sense that the amplitude ratio at the given frequency decreases before the mainshock and increases after, in a quite symmetric fashion. Comparing with the central panel, which is substantially the same of Fig.3, the authors find a similar pattern at a similar frequency for 4<M<5 mainshocks. In this case however the decrease of the amplitude ratio before the mainshock and the subsequent increase after is less pronounced. The same holds for 3<M<4 where the changes of the amplitude ratio are even less pronounced. This is really interesting since it suggests that you can correlate the slope of the amplitude ratio (at a specific frequency) with the magnitude of the incoming mainshock. I invite the authors to focus on this very important result and I suggest some checks to support the scenario. i) I don't fully understand the smoothing procedure: "The common-mode vibration is sliced ....". The really important point is that the amplitude ratio plotted at time t only contains waveforms recorded up to time t. In other words, it is fundamental that quantities evaluated before the mainshock are not contaminated by the mainshock signal. ii) The authors use the signal from 33 seismometers. What happens if I consider a smaller number? In particular how much results depend on the distance between the seismometer and the mainshock? iii) There is some reason to take the first 20 principal components. What happens if one

changes this number? iv) Is there any pattern observed for a single M4+ earthquake, without stacking their signals?

3) I am not totally convinced that the mechanism of resonance is the one responsible for the above observation. In my opinion this is a weaker point which can be also moved to supplementary, keeping a small discussion in the text.

Summarizing, I believe that the direct analysis of seismic waveforms can contain more information than the one extracted from seismic catalogs. This is for instance shown in recent publications (Lippiello et al. Geophys. Res. Lett. and Lippiello et al. Nature Communications 2019). In this direction, the PCA method used by the authors is very promising. I invite the authors to a global rewriting of their manuscript in order to better stress the main results. I also invite the authors to perform the suggested or similar checks to support their findings.

---

## Author Comment (AC1) · 10 Jul 2020

The authors of the submitted research analyse with mathematical/statistical tools published seismic event catalogues from areas of high seismicity (Taiwan and Japan) in an attempt to identify patterns in the distribution on time and space of foreshocks of larger events. The presented results point to a distribution much wider of the foreshocks in time (up to 60 days) and space (up to 400 km of the main shock epicentre) of those currently accepted, even for main shocks of moderate magnitude. Such kind of analysis is promising; but I think as performed and presented in the submitted research is not yet ready for publication. To me, it looks like the pieces of the submitted paper have been assembled in a hurry. The used methodologies need more explanation (why and how they are applied). Even more comments on the choice of the data are also needed. Moreover, a revision of the English syntax is needed. The sense of phrases is difficult to follow in many cases.

[Figure]

For these reasons I think the submitted research needs a deep and throughout revision before it can be accepted for publication. In the following paragraphs I point some specific questions to be addressed on the submitted text.

-Methodology- Line 85. Citation Chen (2014) is not in the reference list.

Reply:

The correct reference is Chang (2014) and has been cited in the manuscript. (Line 96)

Lines 87-89. It is necessary to introduce a minimum description on how ZMAP software removes aftershocks.

Reply:

" The ZMAP software package for MATLAB (Weimer, 2001) was utilized to remove and/or omit influence from duplicate events, such as aftershocks." has been rewriten as "To distinguish dependencies from independent seismicity, the earthquake catalogs are declustered. Therefore, the ZMAP software package for MATLAB (Weimer, 2001) was utilized to remove and/or omit influence from duplicate events, such as aftershocks. The declustering algorithm used in ZMAP is based on the algorithm developed by Reasenberg (Reasenberg, 1985)." (Lines 98-102)

Lines 89-95. Idem: a minimum description on how clusters are classified and the meaning of the input parameters is necessary.

Reply:

" We classify clusters by using the standard input parameters (proposed in Reasenberg, 1985 and Uhrhammer, 1986) for declustering algorithm. The minimum and maximum values of the look-ahead time for building clusters are 1 and 10, respectively. The probability of detecting the next clustered event used to compute the look-ahead time is 0.95. The effective minimum magnitude cut-off for catalog is given by 1.5 and the xk factor for the increase of the minimum cut-off magnitude during clusters is given by

0.5." has been rewriten as "We classify clusters by using the standard input parameters (proposed in Reasenberg, 1985 and Uhrhammer, 1986) for the declustering algorithm. Because the aftershock clusters in a small area and in a short period of time do not conform to the Poisson distribution, which requires removing the aftershocks from the earthquake sequence. Therefore, some parameters can be set as follow: The look-ahead time for un-clustered events is in one day, and the maximum look-ahead time for clustered events is in 10 days. The measure of probability to detect the next event in the earthquake sequence is 0.95. The effective minimum magnitude cut-off for the catalog is given by 1.5, and the interaction radius of dependent events is given by 10 km (van Stiphout et al., 2012)." (Lines 103-111)

Line 95. "The 10 of crack radii: : :" Do you mean 10 times the crack radii? Please, make clear this phrase.

Reply:

Sorry for the ambiguous statement, the sentence is indicated as the interaction radius of dependent events is given by 10 km. The modified description is listed at lines 110-111.

Line 96. Cite Stiphout (2012) is missing in the reference list.

Reply:

The reference has been added in the list.

van Stiphout, T., J. Zhuang, and D. Marsan (2012), Seismicity declustering, Community Online Resource for Statistical Seismicity Analysis, doi:10.5078/corssa52382934. Available at http://www.corssa.org.

Lines 99-102. If I understand properly "crack" and "break" events are definitions you introduced in your analysis, being "crack events" quite equivalent to foreshocks and aftershocks. Please, make clear all these terms.

Reply:

Sorry for the confusion. The "crack" and "break" events have been defined in the manuscript in lines 114-125. Note that we assumed the break event is an earthquake. The crack events can be foreshocks and aftershocks. We stack the crack events to the break events by the time and spatial distance to examine their relationship.

Lines 102-104. There is some problem with the minimum completeness magnitude of the catalogues. Looking at figures S1-S4 it looks like the events in the Taiwan catalogue are included in the Japanese catalogue. Something should be said about this fact. Moreover, the Japanese catalogue comprises many events far away from the main islands (23-34N, 138-147E). I think this whole region does not have the dense seismometer network claimed in lines 83-85. All these points should be clarified in the text.

Reply:

Thank you for your comments. We have modified the results of the Japan catalogs by using the earthquakes that occurred in the northern side of the latitude of 32°N to mainly concentrate in areas with the dense seismometer network and to avoid the double counts of earthquakes in the Taiwan catalogs (lines 187-191). This result is consistent with the previous results, but in order to avoid the problems raised by the reviewer, the revised version will be based on this result. (also see Figs. 1 and 2 in the revision)

Lines 109-110. I assume the spatial and temporal resolutions of the grid are a choice of the authors. If so you may comment if you try other resolutions and/or the reasons for your choice.

Reply:

Sorry for the confusion. The statements have been revised as "Note that the spatial and temporal resolutions of the grids of the spatiotemporal distribution are 10 km and 1

day, respectively, based on the declustering parameters in the ZMAP software." (lines 130-132). Note that the statements associated with the declustering parameters also used in the ZMAP software and the declustering process in ven Stiphout et al., (2012) and Zare et al., (2014). (Lines 107-111)

Reference

Zare, M., Amini, H. and Yazdi, P.; Recent developments of the Middle East catalog, J. Seismol., 18, 749–772, 2014. https://doi.org/10.1007/s10950-014-9444-1.

Lines 113-116. The superimposition process statistical tool should be described. It is not a common tool in seismicity studies.

Reply:

The associated statements have been revised and added for clarification. In practice, the superimposition is a process to stack numerous datasets that can migrate unique features for a few datasets and enhance common characteristics for the most datasets. The count in each grid of the spatiotemporal distributions for all the break quakes are superimposed as a total one based on the occurrence time and epicentral distance of the break quakes. (Lines 138-143)

Lines 118-121. It is not clear to me what "migrate rare characteristics" means. Please clarify this phrase.

Reply:

The associated statements have been revised as "In practice, the superimposition is a process to stack numerous datasets that can migrate unique features for a few datasets and enhance common characteristics for the most datasets.". (Lines 138-140)

-Analytical Results- Lines 130-132. All M2 events are foreshocks or aftershock of M3 events? Cannot they be independent events?

Reply:

This is a very interesting comment. Initially, the opinion from the authors are the same with the reviewer. M2 events can be foreshocks or aftershock of M3 events. Meanwhile, the authors understood that the ZMAP may not fully remove the influence from M2 aftershocks. In fact, we analyze the data without any assumption except for taking the break event as an earthquake. Here, we made the artificial events as the break events for the tests based on that the relationships between the artificial and break events are (1) independent in the time and spatial domain; (2) time dependent (i.e., the same occurrence time but distinct location); (3) location dependent (i.e., the same occurrence location but distinct time). These results are processed by using the same method to construct the spatiotemporal seismic density distributions and the spatiotemporal normalized variations in Figs. A and B (in below) for comparison. No significant increase of the seismic activity can be observed in Figs. 1b-d and 2b-d for the artificial events. In contrast, we can find increase of the seismic activity in Figs. 1a and 2a. This suggests that M2 events could related to the M3 events with a variable distance along the time. (Lines 161-175).

Lines 132-134. What does it means that S/N ratio increases 135 times? Please clarify.

Reply:

The associated statement has been removed.

Another issue: 17993 M3 events in the period 1991-2017/6 mean 2 events per day roughly. As Taiwan is 400 km long approx., it means that in a period of 60 days and 400 km as you are using in your analysis. there are many M3 earthquakes (100 approx.). It is not clear to me how the M2 events are associated with the M3 events. Maybe a good description of the superimposition process as applied in this case clarifies this issue.

Reply:

Thank you for your comments. Those M3 events do not occur in the same region. Instead, the M3 events widely distributed in Taiwan and Japan areas. The distances

between the M2 and M3 events are utilized as an important parameter in this study. The distances from the M2 to the distinct M3 events are different. This suggests if the relationship does not exist that can be mitigated through the stacking processes due to the distinct spatial distribution dominated by the diffident distances (also see Figs. A and B). In contrast, if the relationship dose exist, it can become obvious after the stacking of more than 10 thousand of the M3-4 events. The authors have rewritten the statements associated with the superimposition process in lines 138-143.

Lines 145-164. The previous pointed issues make difficult to follow the discussion on the results.

Reply:

Sorry for the confusion. In this paragraph, the authors focus on the areas with the increase of seismicity density before earthquakes that extends from the fault rupture zone to an external place. The associated statements have been written in the manuscript (Lines 175-212).

-Discussion- In fact this section presents a different analysis, using seismograms and the PCA method. Certainly, the presented analysis has been inspired by the results obtained in the previous section; but it can be performed and presented in a totally independent form. Thus, it should be better presented as another section of analysis results. It is not clear how you are using the PCA analysis in this case. Some figure showing an example of the procedure described on lines 217-222 can help.

Reply:

The statements associated with the descriptions and results associated with the PCA have been move to the new section of the principal component analysis (PCA) on the continuous seismic waveform in lines 234-248. Figure 3 has been added to reveal the energy of the principal components and the first principal component retrieved from continuous seismic waveforms at 31 stations. Figure S5 have been moved to the main

text as Figure 4. Note that Figure 4a shows that the amplitude ratio associated with the Meinong earthquake without the superimposition or the stacking process. For the superimposition or the stacking results associated with the M4-5 and M3-4 earthquakes are shown in Figs. 4b and 4c.

Lines 237-246. There are a lot of suppositions on the used dimensions. If horizontal dimensions (100 x 100 km2) can be roughly deduced/assumed from the previous results (obtained in this section and the previous one), the thickness between 500-1000 m needs a good explanation.

Reply:

The authors appreciate that the reviewer can accept the area in the horizontal dimensions. If the resonance model in the manuscript is true, the unknown parameter of the stress plate is the thickness. The area in the horizontal dimensions is given by the observation in this study. The resonance frequency is obtained by the results of continuous seismic waveforms. The thickness between 500-1000 m is obtained based on the resonance model. The authors just propose a potential model to connect the wide area of increase of seismic activity and the frequency characteristics of crustal vibrations. The authors do not have any evidence to support the thickness between 500-1000 m. In fact, the thickness of the seismogeneric areas is smaller than it of the crust that can be one of the candidates of potential causal mechanism. The authors understood that the debate of the resonance model cannot be solved immediately. We have shortened the statements associated with the resonance model. The original Fig. 4 has been moved to the supplementary for references.

Lines 275-276. I cannot see the need for this citation here. Even more, I has been unable to find the value 2700 km/mˆ3 in the cited paper or on the additional information.

Reply:

The reference has been removed from the manuscript.

Figure Caption

Fig. A. Spatiotemporal seismic density distributions in Taiwan. (a) is computed by the M2 events relate to the real M3-4 events. (b) is computed by the M2 events related to the random M3-4 events in the time and frequency domain. (c) is computed by the M2 events related to time dependent M3-4 events. (d) is computed by the M2 events related to location dependent M3-4 events.

Fig. B. Changes of spatiotemporal normalized variations in Taiwan. (a) is computed by the M2 events relate to the real M3-4 events. (b) is computed by the M2 events related to the random M3-4 events in the time and frequency domain. (c) is computed by the M2 events related to time dependent M3-4 events. (d) is computed by the M2 events related to location dependent M3-4 events.

[Figure]

**Fig. 1.** Fig. A. Spatiotemporal seismic density distributions in Taiwan.

[Figure]

**Fig. 2.** Fig. B. Changes of spatiotemporal normalized variations in Taiwan

---

## Author Comment (AC2) · 10 Jul 2020

The manuscript presents results which, in my opinion, can be very relevant for the forecasting challange. However I find that they are not well presented and the discussion appears quite confusing for the following reasons:

The first part of the manuscript is devoted to study spatio-temporal patterns of seismic activity before and after events in a given magnitude range, for Taiwan and Japan. There are many papers which report a similar increase of seismic activity before large earthquakes. The key point is if the observed increase can have a prognostic value or it can be explained within normal aftershock triggering.

Reply:

The authors fully agree with the comment. The results for the increase of seismic

activity close to the epicenter observed in Figs. 1 and 2 are consistent with the observation in the previous studies (Lippiello et al., 2012, 2017, 2019; de Arcangelis et al., 2016). The associated statements have been added in the manuscript (lines 171-174). The agreement suggests that the increase of seismic activity in Figs. 1 and 2 is not contributed by aftershocks but a prognostic value.

I just suggest some papers where this point is detailed discussed, other references can be find therein (Lippiello et al., Scientific Reports 2012, de Arcangelis et al. Physics Reports 2016, Lippiello et al., Pure and Applied Geophys. 2017, Lippiello et al., Entropy 2019).

Reply:

Thank you for the suggestions. The authors have added those references in the manuscript for intensely supporting our results (lines 171-174). Meanwhile, we are glad to find that the similar pattern (i.e., sudden increase and gradual decrease of the seismic density before and after the earthquakes) can be confirmed by using the different method.

In my opinion many of the results of sec.3 are not really interesting since they are probably artifact of the adopted stacking procedure. Furthermore they are not strictly related to what for me are the main findings (see my point 2). Therefore, I believe that this section can be moved to the supplementary materials whereas in the main-text the authors can just summarize some results and discussing recent literature on this specific point.

Reply:

Thank you for the comments. The authors have shortened the statements, which is similar with the observation in the previous study (lines 153-174). In fact, the manuscript focuses on the increase of seismicity density before earthquakes that extends from the fault rupture zone to an external place. The associated statements have

been extended and added in the manuscript (lines 175-212).

2) Conversely, I strongly encourage the authors to move fig.S5 from the supplementary to the main-text. I am really impressed by this figure. In particular I find striking the result of the left panel which, if i correctly understand, is for a single M6.6 mainshock and therefore is not contaminated by spurious effects caused by the stacking procedure. This figure shows a change in the dominant frequency from roughly $10^{-4}$Hz up to 30 days before, to a much larger value before the mainshock.

Reply:

Thank you very much. Fig. S5 has been moved from the supplementary to Fig. 4 in the main text.

What I find really interesting is the analysis at a fixed frequency (around $10^{-4}$Hz) as function of the time from the mainshock. In this case you find that the mainshock occurrence time is a minimum" point in the sense that the amplitude ratio at the given frequency decreases before the mainshock and increases after, in a quite symmetric fashion. Comparing with the central panel, which is substantially the same of Fig.3, the authors find a similar pattern at a similar frequency for 4<M<5 mainshocks. In this case however the decrease of the amplitude ratio before the mainshock and the subsequent increase after is less pronounced. The same holds for 3<M<4 where the changes of the amplitude ratio are even less pronounced. This is really interesting since it suggests that you can correlate the slope of the amplitude ratio (at a specific frequency) with the magnitude of the incoming mainshock. I invite the authors to focus on this very important result and I suggest some checks to support the scenario.

Reply:

Thank you very much.

i) I don't fully understand the smoothing procedure: "The common-mode vibration is sliced ....". The really important point is that the amplitude ratio plotted at time t only

contains waveforms recorded up to time t. In other words, it is fundamental that quantities evaluated before the mainshock are not contamined by the mainshock signal.

Reply:

Based on the window of 5 days for the slice, the amplitude ratios 5 days before and after earthquakes can be influenced by the mainshock signals. In fact, the enhancements of the amplitude ratios with variable frequency appear more than 20 days earlier than the mainshocks. This suggests that the observed enhancements of the amplitude ratios are not contributed by the mainshock signals.

ii) The authors use the signal from 33 seismometers. What happens if I consider a smaller number? In particular how much results depend on the distance between the seismometer and the mainshock?

Reply:

Based on the pre-earthquake crustal deformation and the numerical model in the previous studies, the seismogeneric areas are considered to be larger than the rupture of the fault zone. In addition, Figs. 1 and 2 also show that the increase of seismicity density before earthquakes that extends from the fault rupture zone to an external place. The radius of the areas with the increase of seismicity density is about 50 km for the M3-4 event and is about 150 km for the M5-6 events. The areas of Taiwan Island are very small. This suggests the signals observed in this study can be recorded in the whole Taiwan island. On the other hand, the upper panel of the Fig. A shows the spatial distribution of amplification ratios in a frequency band between $8 \times 10^{-5}$ to $2 \times 10^{-4}$ Hz for an interval of 0–25 days before the Meinong earthquake. The enhancements roughly cover the whole Taiwan Island. Therefore, the signals can be retrieved from most continuous waveforms from most seismic station. Note that we also take the vertical component of curst displacements from the GNSS data into consideration (the lower panel in Fig. A). An agreement in variations of the spatial distribution of amplification ratios can also be obtained. This suggests that the amplification ratios distribute

in areas with epicentral distance > 250 km. Fig. A has been utilized in the paper that is considered for publication in the other journal.

iii) There is some reason to take the first 20 principal components. What happens if one changes this number?

Reply:

This is a very good question. In the original version, we take the first 20 principal components due to that the threshold of 75% energy is required by other studies. In fact, we can have similar results while the first principal component (12% for energy) is utilized. Note that we have replaced the results in Fig. 4 by using the first principal component in the revision.

iiii) Is there any pattern observed for a single M4+ earthquake, without stacking their signals?

Reply:

Fig. B in below shows the results for a single M4+ earthquake, occurred in the central Taiwan. The enhancements in the frequency between 5x10^-4 Hz and 10^-3 Hz can be found that is in agreement with the observation in the previous study.

3) I am not totally convinced that the mechanism of resonance is the one responsible for the above observation. In my opinion this is a weaker point which can be also moved to supplementary, keeping a small discussion in the text.

Reply:

Thank you for the comments. The associated statements have been reduced (lines 296-310). The associated figure has been moved to the supplementary.

Summarizing, I believe that the direct analysis of seismic waveforms can contain more information than the one extracted from seismic catalogs. This is for instance shown in recent publications (Lippiello et al. Geophys. Res. Lett. and Lippiello et al. Nature

[Figure]

Communications 2019). In this direction, the PCA method used by the authors is very promising. I invite the authors to a global rewriting of their manuscript in order to better stress the main results. I also invite the authors to perform the suggested or similar checks to support their findings.

Reply:

Thank you very much. These important recent publications have been cited in the revision. We have intensely rewritten the more than 50% statements in the manuscript. We have carefully performed the suggested or similar checks to support our findings.

Figure Caption

Fig. A. Spatial distribution of amplification ratios computed from seismic and GNSS data for an interval of 0–25 days before the Meinong earthquake. The upper (a)–(e) and lower (f)–(j) panels denote amplification ratios obtained from seismic and GNSS data. The amplification ratio of > 1 (or < 1) suggests enhancement (or attenuation) of ground vibrations in the particular time period. Time intervals for (a)–(j) indicate distinct time spans until the occurrence of the earthquake during which the data were used for the analysis process. The red star denotes the epicenter. The red lines indicate portions of circles with a radius of 300 km from the epicenter of the earthquake.

Fig. B. The amplitude ratios of the time-frequency-amplitude distribution of one M4.6 earthquake at (121.34E, 23.37N) in the Taiwan region on Dec. 12, 2015.

[Figure]

**Fig. 1.** Fig. A. Spatial distribution of amplification ratios computed from seismic and GNSS data for an interval of 0–25 days before the Meinong earthquake.

**A M4 earthquake**

Frequency (Hz)

$1x10^{-3}$

$5x10^{-4}$

$1x10^{-4}$

$5x10^{-5}$

Time (Days to earthquakes)

-120 -100 -80 -60 -40 -20 0 20 40 60 80 100 120

Amplitude ratio

1.8
1.6
1.4
1.2
1
0.8
0.6
0.4
0.2
0

**Fig. 2.** Fig. B. The amplitude ratios of the time-frequency-amplitude distribution of one M4.6 earthquake at (121.34E, 23.37N) in the Taiwan region on Dec. 12, 2015.

---

## Author Response (AR3)

I have read the revised version of the manuscript and i find it much improved.

I still have some suggestions before publication:

Reply:

Thank you very much.

i) I suggest to present a plot (also as an inset of Fig.4) of the amplitude ratio as function of time at the fixed frequency $10^{-4}$ Hz $\pm 10\%$ for all the three panels of Fig.4. This will show an increase of the amplitude at negative time, a decrease when time goes to zero and an increase later. This plot also allows the reader to better quantify the difference of the amplitude increase as function of the mainshock magnitude;

Reply:

The amplitude ratios as function of time at three fixed frequencies of $1\times10^{-4}$ Hz, $5\times10^{-4}$ Hz, and $1\times10^{-3}$ Hz have been added in Figs. 4d-4f, respectively. An increase of the amplitude at negative time, a decrease when time goes to zero and an increase later can be observed from the ratios at frequency of $1\times10^{-4}$ Hz in Fig. 4d. The amplitude ratios of the enhancements and earthquake magnitudes generally show a proportional relationship.

Associated statements have been added in the revision in lines 283-304.

ii) I am not fully satisfied for the authors'answer about the dependence of results on the number of used seismometers. Indeed it is important to understand if this method can be efficient also in regions with a less dense seismic network.

I therefore invite the authors to perform the same analysis of the new Fig.4 by considering only half of the seismometers (16). It should be not too complicated for the authors.

Reply:

We have reproduced the associated results by using 16 seismometers shown in Fig. A (below).

The enhancements mainly range between $\sim5\times10^{-4}$ Hz and $\sim10^{-3}$ Hz that can be consistently observed in the results using 16 seismometers. Note that the results from 16 seismometers seems clearer due to a removal of noisy stations.

[Figure]

Fig. A. The amplitude ratio of the superimposed time-frequency-amplitude distribution associated with earthquakes with distinct magnitudes using a half number (i.e., 16) of seismometers. The superimposed results 120 days before and after quakes with the M6.6

Meinong earthquake, $4 \leq M < 5$ and $3 \leq M < 4$ are shown in (a), (b) and (c), respectively.

The distribution is normalized for comparison by using the average amplitude in each frequency band of 30 days before and after the quakes. The total number of earthquakes in each magnitude group is shown in the title of each diagram.